# Structured Covariance Modeling Using Learned Mixture-of-Bases for Uncertainty in 3D Segmentation

Peter J.T. Kampen [*,1], Andreas W. Aspe [*,1], Kristine Aa. Sørensen[1,2], Anders N. Christensen[1], Morten R. Hannemose[1], Anders B. Dahl[1], Rasmus R. Paulsen[1], and Josefine V. Sundgaard[1,2]

[1]DTU Compute, Technical University of Denmark, Kongens Lyngby, Denmark
[2]Novo Nordisk A/S, Søborg, Denmark

## Abstract

Accurate segmentation is essential in error-critical domains such as medical imaging, where outputs support clinical decisions. Probabilistic models like the Stochastic Segmentation Network (SSN) enable uncertainty quantification, but existing methods typically use low-rank plus diagonal covariance structures that struggle to capture both global and local spatial correlations, limiting performance gains over deterministic models. We revisit low-rank formulations and introduce two approaches - Single-Basis and Mixture-of-Bases decompositions - that project predicted noise onto learned covariance bases, either globally or within partitioned volume blocks. This yields richer, more flexible uncertainty modeling with minimal parameter overhead. On the most challenging organs in the 3D TotalSegmentator CT dataset, our methods significantly improve Dice scores over deterministic and baseline stochastic models while preserving strong calibration, with the Mixture-of-Bases performing best. These findings show that basis-driven covariance modeling can enhance segmentation accuracy and uncertainty estimation in 3D medical imaging.

## 1 Introduction

Uncertainty quantification has become a central research topic in deep learning-based image segmentation [1]. While deterministic models often achieve strong performance, they provide only point estimates and neglect predictive uncertainty. This can lead to overconfident errors, as modern neural networks are often poorly calibrated, meaning that output confidences do not reliably reflect true probabilities [2]. In clinical imaging applications, incorrect segmentations can have severe consequences, as they may mislead professionals without providing any indication of doubt. This is especially important in organ segmentation, where inaccurate delineation of anatomical structures can compromise treatment planning or downstream quantitative analyses [3, 4].

Automated delineation on Computed Tomography (CT) is particularly challenging in regions where boundaries are faint or anatomy is complex [5], often resulting in overconfident predictions. By explicitly modeling uncertainty, segmentation models can highlight regions of low confidence, enabling clinicians to interpret results more cautiously and reduce the risk of diagnostic errors. At the same time, improved uncertainty modeling may further boost segmentation accuracy through its inherent regularizing properties. Furthermore, quantifying uncertainty reduces the "black-box" nature of deep learning, making AI models more trustworthy and accepted by professionals in other fields [6–8].

Various approaches have been proposed to capture predictive uncertainty in segmentation tasks. Monte Carlo Dropout is widely used to approximate Bayesian inference due to its simplicity, although its accuracy has been questioned [1]. Generative models such as the Probabilistic U-Net and diffusion-based methods introduce latent variables to generate multiple plausible segmentations [9]. Since the introduction of Stochastic Segmentation Networks (SSNs) [10], it has become increasingly popular to model spatially correlated uncertainty in the logit space using a multivariate normal distribution where the covariance is decomposed into a diagonal and low-rank contribution. However, this parameterization has been shown theoretically and in simulated settings to have certain deficiencies [11]. Despite this, subsequent work has continued to largely rely on the low-rank covariance parameterization introduced in the original SSN paper. In contrast, block-based approximations, where the covariance is divided into smaller independent blocks, have demonstrated more robust behavior in spatial domains [11].

In this work, we revisit the low-rank covariance parameterizations and propose an improved formulation. Specifically, we introduce a framework that enhances low-rank modeling through learned basis representations and block-wise decomposition. This hybrid approach enables expressive and computationally efficient uncertainty estimation for volumetric segmentation, capturing both local and global spatial correlations more accurately than standard low-rank models. In summary, our contributions are:

---

[*] These authors contributed equally.
[†] Code available at: https://github.com/andreasaspe/mob-seg3d

Proceedings of the 7th Northern Lights Deep Learning Conference (NLDL), PMLR 307, 2026.

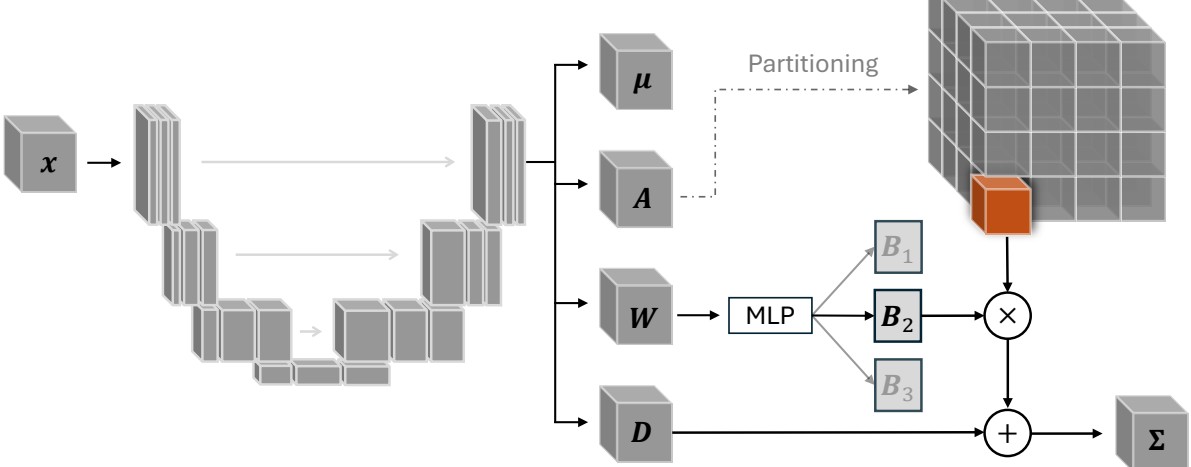

**Figure 1.** Overview of our proposed method. $x$ denotes the input 3D volume and $\mu$ denotes the mean prediction, similar to the standard nnU-Net output. $A$ is the predicted noise structure for each voxel with rank $R$. $W$ represents the weighting of the learned bases $B_1$, $B_2$ and $B_3$. These are multiplied with a partitioned version of $A$ and added with $D$, which accounts for the diagonal variance contribution. Together, this models the covariance matrix $\Sigma$, which, along with $\mu$, defines a normal distribution over the network's output logits.

1. We propose a stochastic segmentation model that projects high-dimensional representations of uncertainty onto a learned basis for improved segmentation performance.

2. We extend this concept to a Mixture-of-Bases (MoB) setting, where bases are allocated spatially according to a weighting scheme.

3. We introduce a principled training framework with KL regularization, orthogonality, Dice- and entropy-based losses that encourage diverse bases, enabling stable learning and robust uncertainty estimates.

4. We demonstrate how these frameworks can improve segmentation performance while providing useful uncertainty estimates on challenging 3D medical imaging segmentation tasks.

## 2 Related Work

Monteiro et al. [10] introduced Stochastic Segmentation Networks (SSNs), which directly model spatially correlated uncertainty in the logit space. In this framework, per-pixel predictions are assumed to follow a multivariate normal distribution. However, since the covariance matrix scales quadratically with the number of pixels and classes, SSNs approximate it using a diagonal plus low-rank decomposition. The mean, low-rank component, and diagonal variance are all predicted by the network.

Zepf et al. [12] extended the SSNs by incorporating Laplace approximations of the posterior over the network weights [8], while retaining the same low-rank covariance structure. The authors develop a fast diagonal Hessian approximation, which has been shown to scale for large neural networks with skip connections.

Recently, Müller et al. [13] proposed a fusion-based approach that combines dropout-based segmentation heads with Laplacian uncertainty estimates. Their method leverages large foundational models to provide image embeddings for downstream segmentation.

Despite these advances, all of the above methods rely on the same low-rank covariance parameterization introduced in [10]. While this structure is computationally efficient, it is too restrictive to capture both local and global dependencies. This limitation is well-documented in spatial statistics, where Stein [11], through an analysis of the eigenvalues of the covariance matrix, demonstrates that restricting the rank to R may capture large-scale variation effectively, but tends to miss fine-scale spatial details. This effect is especially pronounced in cases where neighbouring observations are very correlated [11], as is typical in spatial data such as 3D volumes. Furthermore, Stein shows analytically and numerically that an independent block-based covariance matrix often provides a much better approximation to the likelihood than a low rank approximation [11].

## 3 Methods

In segmentation, uncertainty is often modeled with a heteroscedastic noise assumption, where the network predicts voxel-wise variances in addition to logits. This means that the amount of noise, or uncertainty, can change between different regions of the volume, rather than being fixed everywhere [14].

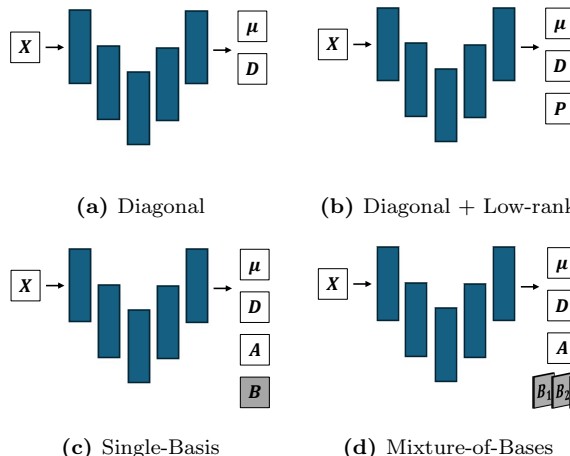

**(a)** Diagonal      **(b)** Diagonal + Low-rank

**(c)** Single-Basis      **(d)** Mixture-of-Bases

**Figure 2.** Comparison of parameterizations evaluated in this study. Panels (a) and (b) illustrate baseline approaches [8, 10], while panels (c) and (d) correspond to our proposed methods. The matrices $B$ are shown in grey to indicate that they are learned parameters rather than input-dependent.

In a standard 3D segmentation task, we consider models that map volumes $\boldsymbol{x}$ into their target volume $\boldsymbol{t}$, $f : \mathbb{R}^{H \times W \times D} \to \mathbb{R}^{C \times H \times W \times D}$, where $C$ denotes the number of classes. We denote the logits as $\eta$. Then the voxel class probabilities become

$$p(\boldsymbol{C}|\eta) = \sigma(\eta) \tag{1}$$

where $\sigma$ is either a sigmoid or softmax function for single- and multi-class cases, respectively. In this work, we assume a Gaussian as the functional form of the distribution over the logits, and hence $p(\eta) = \mathcal{N}(\eta|\boldsymbol{\mu}, \boldsymbol{\Sigma})$ [8]. In a world with infinite computing resources, we could model this distribution in full fidelity by learning both the mean $\mu(\boldsymbol{x})$ and the full covariance matrix $\Sigma(\boldsymbol{x})$ with neural networks, i.e.

$$p(\eta|\boldsymbol{x}) = \mathcal{N}\left(\eta|\mu(\boldsymbol{x}), \Sigma(\boldsymbol{x})\right). \tag{2}$$

However, the number of elements in the covariance matrix scales quadratically with the number of voxels in the volume; hence, the formulation is intractable even for small volumes. Instead, we decompose $\Sigma(\boldsymbol{x})$ into a diagonal variance $D$ and a low-rank covariance contribution $P$ [8, 10]

$$\Sigma(\boldsymbol{x}) \approx \alpha \, D(\boldsymbol{x}) + \beta P(\boldsymbol{x})P(\boldsymbol{x})^T, \tag{3}$$

where $P(\boldsymbol{x}) \in \mathbb{R}^{H \cdot W \cdot D \cdot C \times R}$ and $R$ is the chosen rank for the low-rank approximation. $D(\boldsymbol{x}) \in \mathbb{R}^{H \cdot W \cdot D \cdot C}$ is a diagonal matrix stored as a vector. $\alpha$ and $\beta$ are scaling parameters, though these are often not included explicitly when $D$ and $P$ are learned. Since all covariance matrices can be Cholesky factorized, it is evident that as the rank increases, so does the fidelity of the approximation [11]. The rank places a crucial limitation on the shape of Gaussian

approximation. For two logits to be independent, their entries in $P(\boldsymbol{x})$ must be orthogonal. However, since maximally $R$ orthogonal directions simultaneously exist in the span of $P(\boldsymbol{x})$, then spurious covariances will necessarily appear.

We propose a novel approach for decomposing the covariance matrix using neural networks, as illustrated in Fig. 1. The key idea is to decompose the covariance matrix into a matrix $A$, which is then projected onto a set of basis vectors. We consider two variants of this approach: a Single-Basis model and a Multi-Basis model, which we refer to as the Mixture-of-Bases (MoB). All methods evaluated in this paper are summarized in Fig. 2, with our two proposed variants shown in Fig. 2(c) and Fig. 2(d), respectively. For comparison, Figs. 2(a) and 2(b) present the baseline network structures: one assuming a purely diagonal covariance matrix and another employing the decomposition in Eq. (3), referred to as Diagonal and Diagonal + Low-rank, respectively.

We define a network $A$ where $A(\boldsymbol{x}) \in \mathbb{R}^{H \cdot W \cdot D \times R}$, i.e, a matrix of rank $R$. Note that unlike $P(\boldsymbol{x})$, it does not have rows corresponding to the class dimension. Secondly, we define a basis matrix $B \in \mathbb{R}^{C \times R}$, which is learned but does not depend on the input. Using these two as a new low-rank term, while still maintaining a diagonal contribution from $D$, we model the covariance as follows: Let $N = DHW$ and denote by $A(\boldsymbol{x})_n = a_n \in \mathbb{R}^R$ and $D(\boldsymbol{x})_n = d_n \in \mathbb{R}^C$ the low-rank scales and diagonal variances at voxel $n$, respectively. With $B \in \mathbb{R}^{C \times R}$ the learned basis, the covariance for each voxel $n$ becomes

$$\Sigma_n = B \operatorname{diag}(a_n)^2 B^\top + \operatorname{diag}(d_n). \tag{4}$$

We then construct a block-diagonal matrix with each entry along the diagonal being $\Sigma_n \in \mathbb{R}^{C \times C}$, hence

$$\Sigma(\boldsymbol{x}) = \operatorname{blockdiag}(\Sigma_1, \dots, \Sigma_N) \tag{5}$$

This approach enables the model to learn a global covariance basis $B$, onto which we project the predicted uncertainty representation $A(\boldsymbol{x})$. In this way, the predicted uncertainty will be regularized by expressing it along dimensions that the model is already familiar with. These operations define the Single-Basis model.

Representing the distribution $p(\eta)$ as in Eq. (4) has an obvious limitation. Namely, we assume the existence of a basis that can account for the structure, both globally across the full dataset distribution and locally across the whole volume. To account for this, we introduce a Mixture-of-Bases setup (MoB), as depicted in Fig. 2(d). The setup in Fig. 2(c) is recovered as a special case where the number of bases is one. Instead of only having one basis, we instead define a set of bases $B_i \in \mathbb{R}^{C \times R}, i = 1, \dots, M$ where M denotes the number of bases. In addition, to account for local variation within the volume, we partition

the predicted uncertainty $A(\boldsymbol{x})$ into $P \times P \times P$ cubes, of approximately equal size. Recall that the entries in $A(\boldsymbol{x})$ correspond to the size of the original volume but with an additional rank dimension. We then define a mapping $W$ where $W(\boldsymbol{x}) \in \mathbb{R}^{(P \times P \times P) \times M}$ that predicts the probability of each of the M basis matrices for each of the $P \times P \times P$ partitions. Hence, we use $W(\boldsymbol{x})$ to predict which basis matrix $B_i$ is most suited for each partition of the volume $A(\boldsymbol{x})$.

Let $P(i,j,k)$ denote a partition, where $i,j,k = 1,\dots,P$ are the cube indices. Then $A_{P(i,j,k)}$ denotes the elements in $A$ that belong to that partition. Additionally, let $\hat{B}$ be the most probable basis matrix for the partition, retrieved by the $\arg\max$ over the M-dimension of $W(\boldsymbol{x})$ for the appropriate partition indices $(i,j,k)$. The partitioned and multi-basis version of Eq. (4) will then be:

$$\Sigma(\boldsymbol{x})_{P(i,j,k)} = \text{blockdiag} \left(\Sigma_n\right)_{n \in P(i,j,k)}, \quad (6)$$

Where

$$\Sigma_n = \hat{B} \, \text{diag}(a_n)^2 \hat{B}^T + \text{diag}(d_n) \quad (7)$$

The entire MoB-pipeline, including the partitioning of $A(\boldsymbol{x})$ into cubes, is depicted in Fig. 1. In our setup, all the models $\mu, D, A, W$ mentioned above share the same features encoded through a nnUNet [15].

## 3.1 Training & losses

We follow standard practices for segmentation and employ a mix of Dice (Dice) and cross-entropy (CE) loss, weighted by $\lambda_{\text{dice}}$ and $\lambda_{\text{CE}}$ respectively.

To ensure that the variance does not collapse to zero, we assume a prior variance of 1. We enforce this by employing a Kullback-Leibler (KL) term given by

$$\mathcal{L}_{KL} = \text{KL} \left(\mathcal{N}(\mu(\boldsymbol{x}), \boldsymbol{I}) || \mathcal{N}(\mu(\boldsymbol{x}), D(\boldsymbol{x}))\right) \quad (8)$$

where $\boldsymbol{I}$ is the identity matrix and $\mathcal{L}_{KL}$ regularizes the predicted distribution, penalizing deviations from a Gaussian with the same mean but identity covariance.

To ensure variability among basis representations, we include an orthogonality loss. Consider the flattened version of the tensor containing the $N$ bases $\boldsymbol{BF} \in \mathbb{R}^{N \times C \cdot R}$. We normalize and compute the Gram matrix $\boldsymbol{G}$ where $\boldsymbol{G}_{i,j} = \langle \frac{\boldsymbol{BF}_i}{||\boldsymbol{BF}_i||}, \frac{\boldsymbol{BF}_j}{||\boldsymbol{BF}_j||} \rangle$ and impose a loss on the scale of the off-diagonal elements

$$\mathcal{L}_{\text{orth}} = \frac{1}{N(N-1)} \sum_{i,j} (\boldsymbol{G}_{i,j} - \boldsymbol{I}_{i,j})^2. \quad (9)$$

Finally, to ensure the exploration of the bases, we impose a loss on the entropy of the weighting $W$ of the bases, thus

$$\mathcal{L}_{\text{weight}} = \frac{1}{|P|} \sum_{p \in P} \sum_i^N W_{p,i} \ln W_{p,i}, \quad (10)$$

where $P$ is the set of partitions. The combined loss during training then becomes

$$\mathcal{L}(\eta, \boldsymbol{x}, \boldsymbol{t}) = \lambda_{\text{CE}} \, \text{CE} + \lambda_{\text{dice}} \, \text{Dice} + \lambda_{\text{KL}} \mathcal{L}_{\text{KL}} \\ + \lambda_{\text{orth}} \mathcal{L}_{\text{orth}} + \lambda_w \mathcal{L}_{\text{weight}}, \quad (11)$$

with the following parameters: $\lambda_{\text{CE}} = 1$, $\lambda_{\text{dice}} = 1$, $\lambda_{\text{KL}} = 5 \cdot 10^{-4}$, $\lambda_{\text{orth}} = 1$. In the training of the Single-Basis model, we omit the $\mathcal{L}_{\text{weight}}$ term. To ensure properly trained backbones and in the interest of convergence speed, we employ the nnUNet training setup to estimate the feature encoder and initial version of $\mu(\boldsymbol{x})$. After convergence of the base model, we include the various prediction heads and bases $D, A, B, W$ in training. During training, we sample $k = 5$ times from the predictive logit distribution $p(\eta|\boldsymbol{x}, f)$ and compute the loss:

$$\mathcal{L}_{\text{total}} = \frac{1}{k} \sum_{i=1}^k \mathcal{L}(\eta_i, \boldsymbol{x}, \boldsymbol{t}), \quad \eta_i \sim \mathcal{N}\left(\mu(\boldsymbol{x}), \Sigma(\boldsymbol{x})\right) \quad (12)$$

For the Diagonal and Diagonal + Low-rank approximations, sampling is performed using the torch distributions package [16]. For our contributions, we employ the following for each partition $p$ with $|p|$ elements and rank $R$:

$$\eta_p = \mu(\boldsymbol{x})_p + D(\boldsymbol{x})_p \circ \boldsymbol{z}_1 + B_p(A(\boldsymbol{x})_P \circ \boldsymbol{z}_2), \\ \boldsymbol{z}_1 \sim \mathcal{N}(\boldsymbol{0}, \boldsymbol{I}_{c \cdot |p|}), \boldsymbol{z}_2 \sim \mathcal{N}(\boldsymbol{0}, \boldsymbol{I}_{|p| \cdot R}). \quad (13)$$

Where $\circ$ denotes element-wise multiplication. Under the common sampling convention that each noise entry $z_2$ is drawn independently (and independently of $A(\mathbf{x})$), the resulting covariance structure is block-diagonal in the spatial index: logits are correlated within each voxel (the $2 \times 2$ block), but do not covary across different voxels. However, since $A(\boldsymbol{x})$ controls the scale of the covariance, it may enforce spatial structure on the predicted noise.

During inference, we calculate the expectation of the predictive distribution over the predicted logits distribution through Monte-Carlo sampling [14]

$$p(\boldsymbol{t}|\boldsymbol{x}) = \mathbb{E}_{p(\eta|\boldsymbol{x})} [\sigma(\eta)] \approx \frac{1}{k} \sum_i^k \sigma(\eta) \quad (14)$$

$$\eta \sim \mathcal{N}\left(\mu(\boldsymbol{x}), \Sigma(\boldsymbol{x})\right).$$

**Table 1.** Number of scans in training, validation, and test sets for each anatomical structure after filtering.

| Structure | Train | Val | Test |
|-----------|-------|-----|------|
| Pancreas | 648 | 130 | 163 |
| Gallbladder | 524 | 105 | 131 |
| Duodenum | 614 | 154 | 154 |
| Adrenal Gland (L) | 640 | 128 | 160 |

## 4  Data & implementation

To evaluate our method, we used the publicly available TotalSegmentator dataset [17, 18], version 2.0.1. This dataset contains 1,228 computed tomography (CT) scans with ground-truth segmentations of 117 anatomical structures, including organs, bones, muscles, and vessels. The scans were randomly sampled from routine clinical data covering a broad range of pathologies, acquisition protocols, scanner types, and institutions.

We focused our experiments on four particularly challenging organs: the gallbladder, pancreas, and left adrenal gland—the three structures with the lowest Dice scores reported in TotalSegmentator—as well as the duodenum, the poorest-performing structure within the gastrointestinal subgroup [18]. To ensure meaningful evaluation, we excluded scans where the target structure was not visible, resulting in four derived datasets (sizes listed in Table 1). Each dataset was split 80/20 into training and test sets, with the training portion further divided 80/20 into training and validation subsets. We trained separate nnU-Net models for each anatomical structure using the *nnUNetPlannerResEncL* configuration with default parameters; these constitute our Deterministic models. After convergence, we extended nnU-Net to implement the architectures in Fig. 2 and fine-tuned them following Section 3.1. All models used the AdamW optimizer with a learning rate and weight decay of $10^{-4}$, along with gradient scaling and clipping. Training ran for up to 20 epochs with batch size 1, to maximize validation Dice. Data loading, augmentation, and test-time sliding-window inference followed nnU-Net defaults. For our two methods and the Diagonal+Low-rank baseline, we used rank $R = 16$, selected for computational feasibility and consistency with other low-rank approaches such as LoRA [19]. To obtain robust performance estimates, all models were trained five times with different random seeds to compute metric standard deviations.

## 5  Results

To evaluate performance on the four TotalSegmentator segmentation tasks [18], we use two metrics: the Dice score, which measures segmentation accu-

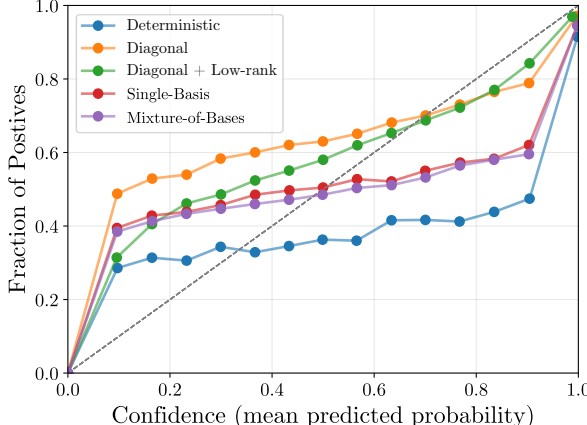

**Figure 3.** Calibration curves for the four stochastic methods and the deterministic model on the pancreas dataset. Sampling is performed uniformly with 15 bins.

racy, and the negative log-likelihood (NLL), which assesses how well predicted probabilities match the ground truth. Table 2 and Table 3 summarize the mean and standard deviation over five runs, comparing the baselines — Diagonal and Diagonal + Low-rank — with our proposed approaches — Single-Basis and MoB.

Table 2 shows that both of our approaches consistently achieve the highest Dice scores, with MoB performing best for three out of four organs. Improvements are most pronounced for pancreas segmentation, where our proposed models significantly outperform the baselines, while still performing better on the remaining organs. In contrast, the baseline models offer little or no gain in Dice score. Standard deviation ranges are comparable across most methods, with the exception of two cases where the Single-Basis model exhibits slightly higher variability.

Table 3 reports the negative log-likelihood results, which reflect a mixture between model calibration and predictive performance. Here, the advantage of our method is less clear. For pancreas segmentation, the MoB still performs best. For duodenum, however, the Diagonal + Low-rank approximation yields significantly better calibration than the other methods. Across all four organs, our proposed models are considerably better calibrated than the deterministic baseline, while also achieving significantly higher Dice scores, thus striking a favorable balance between segmentation accuracy and calibration.

Calibration performance is further illustrated in Fig. 3, which compares model confidence against the fraction of true positives. A perfectly calibrated model would lie on the diagonal. This is measured in Table 4, which reports metrics quantifying deviations from the diagonal, weighted by the number of samples in each probability bin. As reflected both visually in Fig. 3 and from the Maximum Calibration

**Table 2.** Average Dice score over the test sets $\pm$ one standard deviation across 5 random initializations, for the stochastic models and the deterministic backbone from which they were trained. **Bold** denotes best performance.

|  | Pancreas | Gallbladder | Duodenum | AG (L) |
|---|---|---|---|---|
| Deterministic | $80.50 \pm 0.33$ | $79.47 \pm 0.15$ | $81.11 \pm 0.16$ | $87.05 \pm 0.32$ |
| Diagonal | $80.37 \pm 0.38$ | $79.61 \pm 0.13$ | $81.29 \pm 0.28$ | $86.69 \pm 0.24$ |
| Diagonal + Low-rank | $80.30 \pm 0.12$ | $79.56 \pm 0.13$ | $81.74 \pm 0.31$ | $86.80 \pm 0.12$ |
| Single-Basis (ours) | $84.01 \pm 0.89$ | $82.53 \pm 0.52$ | $82.45 \pm 0.34$ | $\mathbf{88.23 \pm 1.22}$ |
| Mixture-of-Bases (ours) | $\mathbf{84.91 \pm 0.32}$ | $\mathbf{82.84 \pm 0.41}$ | $\mathbf{83.06 \pm 0.30}$ | $87.48 \pm 0.35$ |

**Table 3.** Average negative log-likelihood score over the test set $\pm$ one standard deviation across 5 random initializations, for the stochastic models and the deterministic backbone from which they were trained. **Bold** denotes best performance.

|  | Pancreas | Gallbladder | Duodenum | AG (L) |
|---|---|---|---|---|
| Deterministic | $0.27 \pm 0.2$ | $0.17 \pm 0.00$ | $0.36 \pm 0.01$ | $0.022 \pm 0.00$ |
| Diagonal | $0.23 \pm 0.02$ | $0.163 \pm 0.00$ | $0.32 \pm 0.02$ | $0.020 \pm 0.00$ |
| Diagonal + Low-rank | $0.21 \pm 0.01$ | $0.146 \pm 0.00$ | $\mathbf{0.28 \pm 0.03}$ | $\mathbf{0.02 \pm 0.00}$ |
| Single-Basis (ours) | $0.226 \pm 0.01$ | $\mathbf{0.12 \pm 0.00}$ | $0.303 \pm 0.00$ | $0.020 \pm 0.00$ |
| Mixture-of-Bases (ours) | $\mathbf{0.21 \pm 0.01}$ | $0.13 \pm 0.01$ | $0.30 \pm 0.01$ | $0.02 \pm 0.00$ |

Error (MCE) in Table 4, the deterministic model is the least calibrated, exhibiting strong overconfidence. The other approaches exhibit more stable patterns, although our two proposed methods still show a tendency toward overconfidence. Interestingly, their curves appear as a middle ground between the deterministic baseline and the Diagonal + Low-rank parameterization, highlighting how our methods navigate the balance between segmentation performance and calibration. It is further underlined in Table 4 where the MoB model is only surpassed by the classical Diagonal + Low-rank model.

To analyse the effects of varying the number of bases, we test the MoB model with up to five bases on the pancreas dataset. The results of this are seen in Fig. 4. For each number of bases, we train five times with random initializations, following the described training procedure, and measure the Dice score on the test set. The highest mean performance is achieved with three or four bases, with four being slightly higher; however, this comes with a substantially larger variance than the other configurations. For this reason, we opted for three bases as the default choice on all datasets.

In Fig. 5, we present a visual example where the MoB model's segmentation closely resembles the deterministic prediction, but both deviate from the ground truth. The figure shows a 2D sagittal slice of a pancreas segmentation. The predicted variances, $\text{Var}_{p(\eta)}[\sigma(\eta)]$, help identify regions where the model struggles to delineate the organ accurately. Comparing the foreground probability heatmap in Fig. 5(c), computed from the logits in Eq. (1), with the predictive variance heatmap in Fig. 5(b), we observe that the variance signal is markedly higher in the regions where the MoB model produces incorrect segmentations. This region of interest is highlighted in the zoomed-in inset. Most importantly, the variance is relatively low and spatially restricted along the rest of the predicted object boundary, a property not shared by the probabilities.

In Fig. 6, we present another example in a 2D coronal slice, where the MoB model makes a significant correction from the deterministic model. Interestingly, the variance heatmap in the corrected area shows that the MoB model still considers most of the area as uncertain. However, the model has managed to move the segmentation away from the

**Table 4.** Expected Calibration Error (ECE) and Maximum Calibration Error (MCE) over the test set for the five different models. **Bold** denotes best performance.

|  | ECE $\downarrow$ | MCE $\downarrow$ |
|---|---|---|
| Deterministic | $0.017 \pm 0.0$ | $0.432 \pm 0.1$ |
| Diagonal | $0.018 \pm 0.0$ | $0.390 \pm 0.1$ |
| Diagonal + Low-rank | $\mathbf{0.012 \pm 0.0}$ | $\mathbf{0.246 \pm 0.0}$ |
| Single-Basis (ours) | $0.017 \pm 0.0$ | $0.318 \pm 0.1$ |
| Mixture-of-bases (ours) | $0.014 \pm 0.0$ | $0.317 \pm 0.1$ |

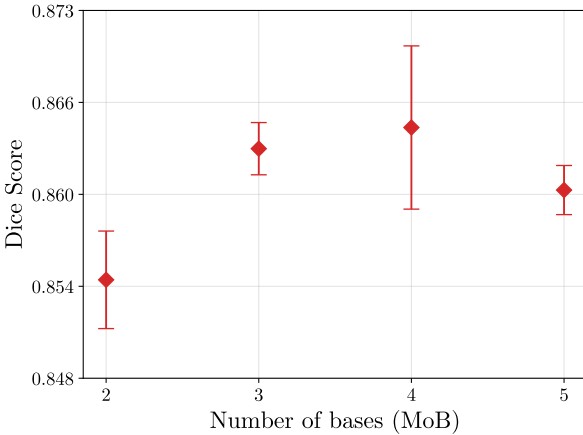

**Figure 4.** Dice score on the pancreas test set as a function of the number of bases in the Mixture-of-Bases model, with whiskers indicating $\pm 1$ standard deviation.

area. The predicted variance, however, indicates that the model is aware that this might be an erroneous segmentation. This too is in line with the results presented in Tables 3 and 4 and Fig. 3.

## 6 Discussion

We compared several covariance parameterizations for uncertainty modeling in 3D segmentation. The Diagonal and Diagonal + Low-rank variants serve as baselines: they improve calibration (NLL, ECE & MCE) compared to the deterministic nnUNet, but bring little to no gains in Dice score. This is expected since the assumption of independence of logits in the diagonal approximation fails to capture the underlying structure, and the added low-rank structure fails to capture the spatial correlations that drive segmentation quality. Moreover, these baselines treat class covariance and spatial covariance at the same level, limiting their ability to disentangle class-specific uncertainty from spatial smoothness.

In contrast, basis-based formulations provide structured covariance with a closer connection to the underlying anatomy. By learning basis functions, we model the class covariance more explicitly, rather than conflating it with spatial correlation. The Single-Basis model already improves Dice significantly, showing that the model can benefit from the added constraint of the forced intra-logit covariance. However, its calibration is less competitive in some cases, reflecting limited flexibility.

Our proposed MoB model addresses this limitation by combining multiple structured components. This yields the strongest and most consistent Dice score performance across segmentation tasks, while maintaining competitive NLL. Importantly, mixtures also reduce variance across runs, suggesting a more stable optimization. These results demonstrate that mixtures of bases strike a favorable bal-

ance between segmentation accuracy and calibrated uncertainty, offering a scalable alternative to purely diagonal or unconstrained low-rank approaches.

We hypothesize that a significant cause for the performance improvement induced by the use of the learned bases originates from implicit noise reduction. In the Diagonal + Low-rank setting, we construct the low-rank component from the predicted $P(\boldsymbol{x})$. However, this prediction naturally contains a level of uncertainty too. If the model fails to predict $P(\boldsymbol{x})$ accurately, the error will propagate directly into the sampled predictions. For our proposed models, a similar problem may be ascribed to $A(\boldsymbol{x})$. However, in this case, $B$ projects the noise prediction, as well as the sampled noise, from $R$ dimensions to two. The dimensionality reduction may serve as noise reduction, as small variations in the higher dimensions may not be present in lower dimensions.

Increasing the number of bases has the potential to further improve performance. The decrease observed in Fig. 4 when moving from four to five bases may result from suboptimal hyperparameter settings, particularly in the loss functions. Importantly, the optimal number of bases is likely tied to the partitioning of $A(\boldsymbol{x})$. In many cases, the ground-truth segmentation occupies only a small fraction of the total volume. Consequently, when $A(\boldsymbol{x})$ is partitioned into blocks, the foreground class often falls entirely within a single block. In such scenarios, one basis can capture the uncertainty of the background blocks (which have minimal uncertainty), while another basis accounts for the foreground. If the partitioning were more fine-grained, multiple subregions of the organ would span several blocks, thereby creating the need for multiple bases to represent this variation.

Overall, our models improve Dice scores across all four organs while providing better calibration than the deterministic nnU-Net. However, their calibration is not always superior to the stochastic baseline (Diagonal + Low-rank). Training with our proposed formulations appears to encourage the model to account for calibration, which in turn enhances segmentation quality. When the MoB model produces an incorrect segmentation, it often signals this by assigning a high predictive variance in the affected region. This uncertainty estimate could therefore serve as a proxy to identify areas that might benefit from post-processing or manual refinement, an aspect not examined in this study. Notably, we evaluated the methods on some of the most challenging organs in the dataset, characterized by ambiguous boundaries and substantial anatomical variability, suggesting that the approach may also benefit other difficult segmentation tasks.

A key limitation of this work is its reliance on data originating from a very homogenous source.

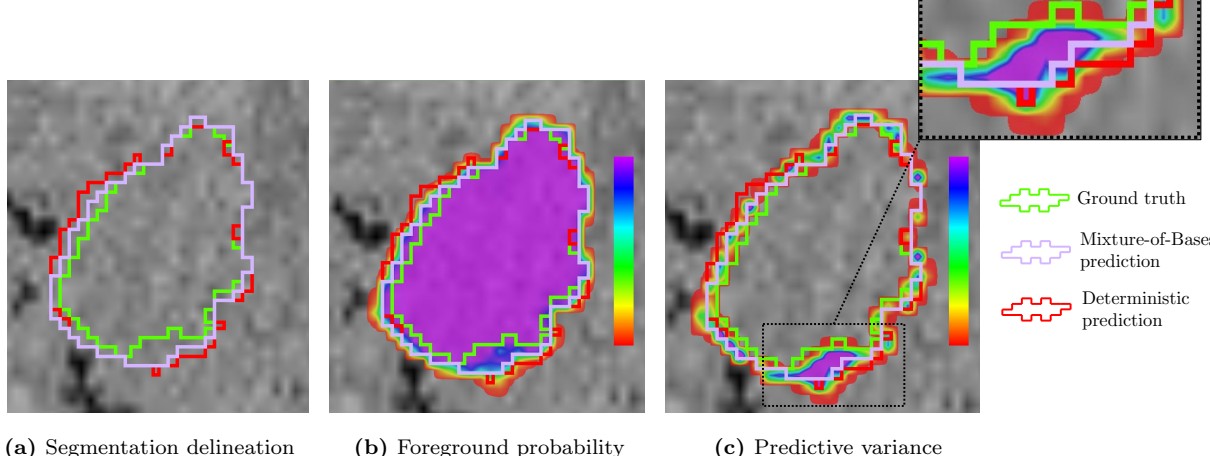

**(a)** Segmentation delineation     **(b)** Foreground probability     **(c)** Predictive variance

**Figure 5.** An example of a MoB segmentation on a 2D sagittal slice where the segmentation fails in certain areas. In (b) and (c), we overlay the probability of foreground and the predictive variance, respectively, using a normalized colorbar.

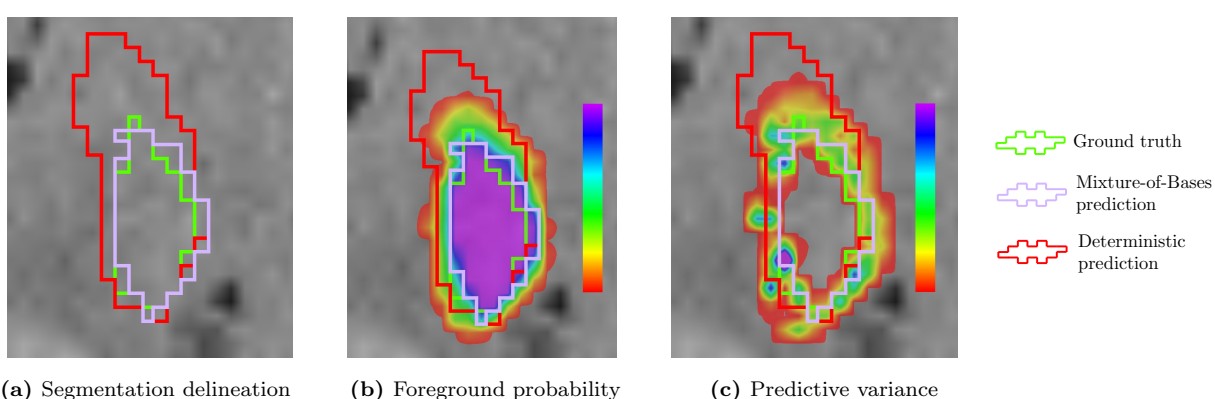

**(a)** Segmentation delineation     **(b)** Foreground probability     **(c)** Predictive variance

**Figure 6.** An example of a MoB segmentation on a 2D coronal slice where the model improves segmentation over the deterministic baseline. In (b) and (c), we overlay the probability of foreground and the predictive variance, respectively, using a normalized colorbar.

Further study into robustness towards distribution shifts is therefore prudent. Additional hyperparameter tuning, perhaps specifically designed towards the target organs, would most likely improve performance. Despite these constraints, the findings serve as a proof of concept, indicating that improved low-rank covariance decompositions can enhance both segmentation accuracy and uncertainty estimation in 3D medical image analysis.

# 7 Conclusion

We investigated alternative covariance parameterizations for uncertainty modeling in 3D medical image segmentation. While baselines that model the covariance as a diagonal plus low-rank term improve calibration relative to deterministic models, they fail to provide meaningful gains in segmentation accuracy due to their inability to capture structured spatial and class-specific correlations. In contrast, our proposed basis-based formulations explicitly disentangle class-specific covariance from spatial variability, resulting in improved segmentation performance while maintaining comparable calibration. Evaluated on the most challenging organs in the TotalSegmentator dataset, the Single-Basis model achieves substantial Dice score gains, while the partitioned Mixture-of-Bases further improves both Dice score and stability, yielding the strongest and most consistent overall performance. Beyond accuracy, the structured covariance also yields interpretable uncertainty estimates, with predictive variance effectively highlighting anatomically ambiguous regions. Taken together, this work serves as a proof-of-concept that basis-driven covariance modeling provides a simple, parameter-efficient, and effective framework for uncertainty-aware segmentation, with strong potential for medical imaging applications where both precise predictions and reliable uncertainty estimates are critical.

# Acknowledgments

Funding for this research was partly provided by Novo Nordisk A/S.

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
