# OpenReview forum: "Structured Covariance Modeling Using Learned Mixture-of-Bases for Uncertainty in 3D Segmentation"
_NLDL.org/2026/Conference — NLDL 2026 Spotlight_

### Official Review · Reviewer_DB8r · 2025-10-03
**Interesting method, interesting but somewhat inconsistent results**

**Rating:** 2
**Confidence:** 2
**Final Rating:** 4
**Final Confidence:** 3

**Summary:**

They study the important issue of error covariance in 3-d segmentation models. They develop a novel extension of established low-rank additions to simple diagonal covariance modelling, where their contribution is to train the model to find one or more linear bases for the additions to the covariance matrix. The math appears sound and it's interesting idea.
They report improvements in segmentation scores, but not in metrics of uncertainty calibration, compared to the low-rank method.

**Strengths:**

The article is well structured and relatively easy to read, despite its advanced technical level, which is good.
The method is well motivated and the math appears sound and correct, as far as I am able to evaluate.
They test over several datasets, and compare their results to standard methods, which is a plus.

**Weaknesses:**

Their novel contribution in the modelling is a more flexible representation of the covariance structure of the error terms. If this were successful, we would definitely expect to see improvements in the NLL compared to the low-rank method, and here we see the opposite. This worries me quite a bit.

On the other hand, they report improvements of segmentation scores, compared to low-rank, and their explanation is that the model understands the error covariance better, which points in the opposite direction of the NLL and ECE/MCE findings.
(I assume the covariance structure is is included in the NLL calculations, which might be mentioned.)
If I read it correctly, the results for the other other methods are all computed by them, and not compared to state-of-the-art for the given problem over-all. This would be appropriate since their positive results are with respect to segmentation accuracy, rather than uncertainty calibration.

I am not a fan of using random variation over training runs with different initializations as a way to construct CIs for performances. This is sort of related to epistemic uncertainty, but not in a convincing way. I would prefer a cross-validation approach to this.

A minor thing: Typo in the Table 4 heading (ECE should be MCE).

**Final Justification:**

They have cleared up the most important issues relating to their focus on NLL.

**Justification:**

My conclusion is mainly based on the fact that the method was designed specifically to improve the error term modelling over the established low-rank approach, which basically failed when the NLL measure did not improve.
I am not absolutely sure of this evaluation, and I may have missed something, so I would be willing to change my mind if they present better arguments.

---

> ### Author Rebuttal · Authors · 2025-10-21
>
> We thank the reviewer for their comments and aim to clarify the main concerns below.
>
> On the scope of the main contribution:
>
> We acknowledge that the framing of our contribution could be clearer. The original motivation was to explore new parameterizations in probabilistic frameworks for 3D segmentation and their effects on uncertainty quantification and segmentation performance. We did this by modeling uncertainty in a structured way, which also allows us to identify regions where the model is less confident. During our investigation, we found that the proposed methodology primarily improves segmentation accuracy while maintaining competitive NLL. We believe this outcome is both scientifically valuable and relevant to the field: Modeling the logit distribution not only aids uncertainty quantification but can also lead to better segmentation quality. We agree that the abstract and introduction may have overemphasized uncertainty estimation, and if accepted, we will adjust the framing to reflect the main contribution more accurately.
>
> On the relation between Dice, NLL, and calibration:
>
> The apparent discrepancy between Dice and NLL is expected and arises from the fundamentally different aspects of model behavior these metrics capture. While Dice reflects the quality of hard segmentation decisions, NLL reflects a combination of calibration and segmentation performance.
> We acknowledge that the phrase “more flexible representation” was too general. We argue that our method provides more flexibility in areas where the intra-voxel covariance affects the error term significantly. Namely in high-uncertainty areas, where the fact that our parameterization does not attempt to account for both intra-voxel and far-reaching spatial covariance structures simultaneously provides the added flexibility. We seek to elaborate below.
>
> In a high-uncertainty setting, where the class logits are similar ($L_0 \approx L_1$ in a two-class case), accurately modeling their covariance before applying the nonlinear softmax can shift the resulting class probabilities $P(C_0), P(C_1)$. This may change which class attains the highest probability (the argmax) and thus correct a previously misclassified voxel, improving segmentation accuracy. However, because the probability difference is small, the impact on the continuous NLL metric remains limited.
>
> In contrast, in a low-uncertainty setting ($P(C_1) \approx 0$ or $P(C_1) \approx 1$), the segmentation decision is already stable. Here, the NLL can vary dramatically if the model is confidently wrong, since it penalizes overconfident errors disproportionately. The NLL, given by $-\log P(\text{TrueClass})$, has no upper bound and increases sharply as $P(\text{TrueClass}) \to 0$. Thus, a few confidently misclassified voxels can dominate the total NLL, even though they have negligible influence on Dice. Adjusting probabilities in such regions can therefore change the NLL substantially while leaving segmentation accuracy unaffected. Hence, a model that moderates overconfidence may obtain better NLL without necessarily improving segmentation.
>
> Since all of the probabilistic models improves NLL over the deterministic model, it is safe to assume that they all to a certain degree moderate overconfidence in low-uncertainty but wrongly segmented areas. We observe an increase in the segmentation accuracy for our contributions indicating that it better captures the underlying distribution in areas where this will affect the segmentation performance (high-uncertainty areas). The fact that we only observe an NLL improvement for the baseline low-rank model and not a segmentation improvement, indicates that it moderates the probabilities more than our methods in low uncertainty but wrongly segmented areas.
>
> Therefore, our methods better capture specific parts of the error distribution — particularly the high-uncertainty regions where accurate covariance modeling most affects segmentation outcomes. The observed discrepancy between Dice and NLL thus reflects an inherent asymmetry between these metrics rather than a methodological weakness. While our models do not always achieve the lowest absolute NLL across all datasets, they remain consistently competitive.
>
> On computing the baseline results ourselves:
>
> It is correct that we computed the results of the baseline methods ourselves. The main reason is that the original implementations of these methods do not employ current state-of-the-art segmentation architectures/training pipelines. To ensure a fair and meaningful comparison, we used the nnU-Net framework as a common backbone, providing a strong deterministic reference model. Baseline implementations were taken from the source code of the original works and placed into our training pipeline accordingly. In fact, the baseline methods are designed to be placed on top of any given model architecture, so we argue that using the same base architecture and training pipeline for all methods is a fair way of comparing.
>
> Regarding comparisons with external state-of-the-art Dice scores, our focus is methodological rather than task-specific optimization. We aim to study the parameterization of the covariance structure, not to fine-tune the segmentation backbone. Using nnU-Net provides a strong and well-established baseline “out of the box,” allowing us to demonstrate that the proposed probabilistic extensions yield consistent performance gains when applied to a high-quality deterministic model. Achieving truly state-of-the-art, task-specific performance would likely require extensive pretraining, task-specific refinement, and the use of larger or more specialized model architectures, which are beyond the scope of this work.
>
> On the use of random restarts for confidence interval creation:
>
> The choice to use random variation as a means of constructing confidence intervals is somewhat divisive, and we understand the reviewer's point. It is related to epistemic uncertainty and the variance induced by the training procedure. We chose to focus the confidence intervals on these sources of variance and hope that the results across the various organs provide sufficient evidence that the performance increase is independent of data variation. The procedure is loosely inspired by Deep Ensembles, but we agree that cross validation is a valid approach as well.
>
> Finally we would like to thank the reviewer for pointing out the typo in Table 4. We will change ECE to MCE in the camery-ready version, if the paper is accepted.
>
> We hope this rebuttal clarified the main concerns and supports the paper’s acceptance at NLDL 2026.

---

### Official Review · Reviewer_x5GA · 2025-10-03
**A solid piece of work with honest and reasonable results**

**Rating:** 4
**Confidence:** 3
**Final Rating:** 4
**Final Confidence:** 4

**Summary:**

The authors present a comparison of low-rank covariance representations for predicting model uncertainty in 3D segmentation. They explore a spectrum of approaches, from deterministic models (no covariance) through low-rank diagonal covariance, to models using multiple bases that capture both global and local structures. The base model is a UNet-like architecture, adapted to output the elements required by each modeling approach. As the authors note, improvements in uncertainty quantification over the baselines are modest, while gains in segmentation accuracy are more pronounced.

**Strengths:**

- The paper is well-written and clearly structured.
- The choice of metrics is appropriate.
- The selection of baselines is sound, covering a logical progression from deterministic models to mixtures of bases.
- The authors provide a thorough discussion of the results and successfully highlight how their proposed mixture-of-bases approach improves segmentation performance by capturing both local and global structures.

**Weaknesses:**

- The study is limited to low-rank modeling.
- The contribution regarding uncertainty quantification is somewhat overstated in the abstract and introduction; most improvements are in accuracy, and, as the authors themselves note, calibration improvements are milder.
- This may be a misunderstanding on my part, but if the output of the default nnUNet-PlannerResEncL includes a confidence measure, is it accurate to call it deterministic?
- The sentence in the abstract from lines 21 to 23 appears incomplete.

**Final Justification:**

Thanks to the author for their answer to my concerns. I'm happy to keep my decision of acceptance of the paper.

**Justification:**

Although the proposed approach does not yield significant improvements in uncertainty quantification—a key aspect of stochastic segmentation—compared to the baselines, the authors present a set of robust experiments and offer an insightful discussion on how capturing both local and global structures can impact segmentation accuracy.

---

> ### Author Rebuttal · Authors · 2025-10-21
>
> We thank the reviewer for their insightful comments. Below we will attempt to answer the questions posed in the review.
>
> On uncertainty quantification vs. accuracy gains:
>
> We appreciate the reviewer’s observation that the improvements in uncertainty quantification are more modest compared to the gains in segmentation accuracy. This is a fair point, which we are fully aware of. We will revise the abstract and introduction to better reflect this balance. While our method does improve calibration metrics relative to the deterministic baseline, the most consistent and significant gains are indeed in segmentation performance.
>
> On the scope of the study:
>
> It is correct that our study focuses on low-rank modeling strategies. Full-rank modeling is infeasible due to the size of the covariance matrix involved. Specifically, we estimate the covariance over all voxels and their respective logits, resulting in a matrix of size $(H \cdot W \cdot D \cdot C) \times (H \cdot W \cdot D \cdot C)$. For a moderately sized volume with $H = W = D = 250$ and $C = 2$, this yields in the order of $10^{14}$ elements. Even with Cholesky factorization, the computational cost remains prohibitive. Therefore, low-rank approximations are a necessary constraint in this domain.
>
> On the deterministic nature of nnUNet-PlannerResEncL:
>
> We thank the reviewer for raising this point. While the nnUNet-PlannerResEncL outputs softmax probabilities, these are point estimates and do not reflect uncertainty in a probabilistic sense. The output is always the same for a given input, and no sampling is involved. In contrast, our formulation models a full distribution over the logits, enabling sampling and the computation of classical uncertainty metrics such as predictive variance. To illustrate: for a function $f$ and input $x$, if $\text{Var}(f(x)) = 0$, the function is deterministic; if $\text{Var}(f(x)) > 0$, the function is stochastic. Therefore, we argue that it is accurate to refer to nnUNet-PlannerResEncL as deterministic in the context of uncertainty modeling.
>
> On the incomplete sentence in the abstract (lines 21–23):
>
> We appreciate the reviewer catching this. The sentence was inadvertently truncated during editing. We will correct it in the final version to ensure clarity and completeness. The sentence will read: "Evaluated on the 3D segmentation task of challenging anatomies from the TotalSegmentator CT dataset, our approaches achieve significant Dice score improvements over deterministic and baseline stochastic models while maintaining competitive calibration, with the Mixture-of-Bases yielding the greatest improvement."
>
> We again thank the reviewer for the comments and recommendation. We hope our answers are satisfactory.

---

### Official Review · Reviewer_CEVP · 2025-10-08
**Solid improvement for stochastic segmentation networks**

**Rating:** 4
**Confidence:** 2
**Final Rating:** 4
**Final Confidence:** 3

**Summary:**

For example, as in typical scenarios such as medical diagnosis and treatment, there are many situations where explicitly evaluating segmentation uncertainty is required in image and 3D image segmentation. Research on segmentation that explicitly captures such certainty has been developing for many years, but since around 2020, Stochastic Segmentation Networks (SSN) has gained attention as a promising direction using neural networks. Subsequently, SSN has seen various developments. These development approaches, similar to the original paper, are based on a strategy that approximates the covariance of spatial correlations using diagonal components and low-dimensional components. However, it has become known that such representations traditionally possess certain vulnerabilities. The authors focus on this point as a research challenge and propose the idea of representing covariance with a block structure and the technique of introducing a mixture model.

**Strengths:**

- The motivation for this paper is explained very clearly from two perspectives. First, it is readily apparent that the accuracy of image segmentation in medical applications must be quantified, as it significantly impacts diagnosis and treatment. Furthermore, from a more specific viewpoint, it is also understandable that this paper aims to resolve a certain flaw in the representation of spatial correlation covariance present in highly promising existing methods.

- The proposed method in this paper is based on a solid improvement over existing state-of-the-art techniques. Representing covariance with a block structure and introducing a mixture model (while not a groundbreaking idea) appears to be a very solid and reasonable improvement in terms of its practical and applied significance.

- The effectiveness of the proposed method is demonstrated very clearly in quantitative terms. To convey its benefits qualitatively as well, the authors have made considerable efforts through two examples. While I am not an expert in medical imaging and cannot adequately judge how significant this improvement is from a medical application perspective, I can see that the AI skillfully resolves a challenge that is very difficult for humans (non-experts).

**Weaknesses:**

- The authors mention the vulnerability of the existing stochastic segmentation networks (SSN) methods in approximating covariance using diagonal components and low-dimensional components. A self-contained, more detailed explanation of what this vulnerability specifically entails would likely be useful to many readers. Indeed, the explanation in the second paragraph of Section 1 and the description in Section 2 make it clear that recent developments in SSN have adopted a modeling strategy similar to the original paper, and that this vulnerability is mentioned in [11]. However, even after reading Section 3, which describes the proposed method, I was unable to understand what specific vulnerabilities exist. Furthermore, after briefly reviewing [11], I remain somewhat unclear on exactly which points readers are referring to. I strongly believe that providing a more concrete description of the vulnerabilities in existing methods (which is also a focus of this paper) would be beneficial to readers.

**Final Justification:**

We appreciate the authors' thoughtful responses.

Our concern (regarding the intuitive explanation of the limitations in the existing study [11]) has been addressed by the authors' refined and concise explanation. Such descriptions are expected to greatly assist diverse readers in correctly understanding the value of this paper.

**Justification:**

This paper presents a very robust improvement to methods equipped with the capability to explicitly evaluate uncertainty in image and 3D image segmentation. Focusing on a shared limitation in SSN and its variants—a prominent approach since around 2020—namely the approximation of covariance using diagonal and low-dimensional components, the authors propose replacing this with a block structure. Comprehensive experiments demonstrate that this improvement yields effective results both quantitatively and qualitatively. Based on the above, I believe this paper is well-prepared for broad discussion within the relevant community.

[Minor comments]

The right sides of Figure 5 and 6 (Mixture-of-bases) may be missing.

---

> ### Author Rebuttal · Authors · 2025-10-21
>
> We thank the reviewer for their positive assessment and constructive feedback.
>
> We agree that an additional - or more elaborate - explanation of the weaknesses of existing approaches would be prudent, and we will include it in the camera-ready if the paper is accepted. These limitations primarily arise from overly restrictive modeling assumptions, which prevent the covariance from approximating the true underlying probability distribution. Below, we briefly summarize the key points.
> - Diagonal:
> This model assumes that all voxels and logits behave independently and assigns each a variance. While this lowers the negative log-likelihood relative to the deterministic model (as seen in Table 1), it cannot capture intra-logit or spatial covariance, and therefore does not improve segmentation accuracy or calibration.
> - Diagonal + Low-Rank:
> This is an improved parameterization, which attempts to model voxelwise correlations through a low-rank matrix $P \in \mathbb{R}^{(H\cdot W \cdot D \cdot C)\times R}$. However, the formulation cannot freely assign independence between distant voxels: rows of $P$ must be orthogonal for independence, and in an $R$-dimensional subspace (here $R{=}8$), only $R$ orthogonal directions exist. This leads to spurious long-range correlations and limited segmentation improvements.
>
> Relation to reference [11] (Stein, 2014):
> Stein further supports the above points by theoretically as well as experimentally showing how low-rank covariance approximations may lead to a lower segmentation performance. This is explained by examining the eigenvalues of the covariance matrix: limiting the rank to R may capture large-scale variation effectively, but tends to miss fine-scale spatial details. The authors argue that this limitation is especially pronounced when neighboring observations are highly correlated, which is exactly the case for most 3D images. In our work we also see qualitatively (Figure 5 and Figure 6) that the segmentation performance indeed is increased due to a better understanding of fine-scale details. Furthermore, Stein demonstrates that when random variance is small, block-based covariance models more accurately represent spatial structure, which is also supported by the improved segmentation results in our work.
>
> We will incorporate a concise version of these explanations in the final paper to clarify the motivation behind our approach.
>
> Minor Comment – Figure 5 and 6:
> We thank the reviewer for pointing this out. We will ensure that the full width of Figures 5 and 6 is clearly visible in the final version.
>
> We again would like to thank the reviewer for their comments and for the positive evaluation of our paper.

---

### Meta-Review · Area_Chair_Ymms · 2025-10-31

**Recommendation:** Accept (Poster)
**Confidence:** 3

**Metareview:**

The presented method for predicting model uncertainty in 3D segmentation is technically sound, experimentally validated, and clearly presented. The significant accuracy improvements combined with maintained calibration quality represent progress for uncertainty-aware medical image segmentation. The reviewers have mostly positive views of the paper, with some weaknesses pointed out that the authors have responded to in the rebuttal.

---

### Decision · Program_Chairs · 2025-11-05

**Decision:**

Accept (Spotlight)

**Comment:**

We recommend an oral and a poster presentation given the AC and reviewers recommendations.

A spotlight presentation refers to a poster selected for an oral highlight but not designated as a full oral presentation per the AC’s recommendation.